# Transition Period and Young Adulthood in Patients with Childhood Onset Growth Hormone Deficiency (COGHD): Impact of Growth Hormone Replacement on Bone Mass and Body Composition

**DOI:** 10.3390/ijms251910313

**Published:** 2024-09-25

**Authors:** Mirjana Doknic, Marko Stojanovic, Aleksandra Markovic

**Affiliations:** 1Neuroendocrine Department, Clinic for Endocrinology, Diabetes and Metabolic Diseases, University Clinical Center of Serbia, 11000 Beograd, Serbia; markostoj@yahoo.com; 2Faculty of Medicine, University of Belgrade, 11000 Belgrade, Serbia; 3Department of Endocrinology, Internal Medicine Clinic, University Clinical Centre of the Republic of Srpska, Faculty of Medicine, University of Banja Luka, 78000 Banja Luka, Bosnia and Herzegovina; dr.aleksandra.markovic@gmail.com

**Keywords:** transition period, GHD, GH therapy, bone, body composition

## Abstract

The aim of this review article is to highlight the consequences of COGHD after the end of linear growth on bone mass and body composition and the opposing beneficial effects of continuing GH replacement in the transition period and young adults. The role of growth hormone in the period of late adolescence and young adulthood is well established, mainly in achieving peak bone mass and a favorable body composition, characterized by muscle mass increase and fat mass reduction. Patients with childhood onset growth hormone deficiency (COGHD), after reaching the adult height, have a reduced bone mineral density and muscle mass with increased fat mass compared to healthy controls. Inadequate body composition is a predictor for cardiovascular risk, while low bone mass in early youth hallmarks the risk of osteoporosis and bone fractures in later life. Cessation of growth hormone replacement (GHr) after completion of growth will lead to delayed peak bone mass and unbalanced body composition with increased abdominal fat deposits. According to numerous clinical studies monitoring the effects of GH treatment on the physical and psychological status of patients with persistent GHD after completion of growth, we suggest continuing this treatment between 16 and 25 years of age. It is advised that GHr in the transition period be administered in intermediate doses between those for the pediatric population and those for the adult population. Usual daily GHr doses are between 0.3 and 0.5 mg but need to be individually optimized, with the aim of maintaining IGF-I in the age-specific normal range.

## 1. Introduction

This narrative review is dedicated to the clinical characteristics of childhood onset growth hormone deficiency (COGHD) in terms of bone and body composition in the period of transition from childhood to adulthood as well as the effects of growth hormone replacement (GHr) on these two systems of young adults aged 16–25 years. The transition period (TP) is defined as the age span starting at the individual point of reaching the adult height and lasting up to complete somatic and psychological maturation. It refers to a wide spectrum of physical and psychosocial changes from late puberty to full adult maturity and spans approximately 6–7 years after the end of growth in height [1,2]. A recent expert study has more clearly defined the TP as the period between the end of puberty and the achievement of peak bone mass (PBM), covering the interval from early adolescence to young adulthood [3]. The authors consider that the transition phase begins when a patient reaches Tanner stage 5 (most often 14.7 ± 2.2 years in boys or 14.0 ± 2.4 years in girls) and lasts to a mean age of 23 years in males and 20 years in females [3].

Until three decades ago, GH replacement was available only for children with a confirmed deficiency of this hormone, and the duration of its administration was limited by linear growth termination. The administration of GH in adults with growth hormone deficiency (GHD) originated in 1989, but it was officially approved in 1996 [4]. Since then, in the next 15 years, there has been a growing body of literature describing the importance of GHD effects on muscle, fat and bone mass in that age group. However, after a period of pioneering findings on the effects of GHr in young adults, there are very few original papers and comprehensive review articles on this topic. Although the positive effects of GHr are evident in the TP regarding bone and muscle health, the cardiovascular system, metabolism and psychological wellbeing, the importance of continuing GHr after reaching the adult height (AH) is still under-recognized or neglected in many countries [5,6,7]. In order to optimize the process of transitioning from pediatric to adult endocrinology care, numerous clinical guidelines have been published that highlight the importance of continuing GHr after the end of longitudinal growth [8,9,10,11,12]. Despite the availability of these algorithms facilitating the transition process, a considerable portion of patients remain without an appropriate endocrine follow-up after the cessation of growth. After reaching near AH, upon transition to adult endocrinology care, young patients treated with GHr in childhood need to be reassessed before a decision is made on further treatment. COGHD patients are to be retested after at least 1 month of GHr interruption. Upon retesting with adult cut-off criteria, approximately two-thirds of patients with isolated idiopathic GHD in childhood qualify as GH sufficient [8]. In patients with organic (tumor or irradiation), structural (hypothalamic or pituitary malformations) or confirmed genetic causes of GHD, reversibility of GHD is rarely expected, and IGF-1 levels below −2 SD along with deficiencies in all other pituitary axes are adequate to confirm GHD persistence without retesting [9]. Adolescents with childhood-acquired GHD (due to a tumor, cranial irradiation or traumatic brain injury) with at least one additional pituitary axis failure require one GH stimulation test to confirm GHD persistence. Patients with isolated GHD require failure in two different GH stimulation tests to confirm GHD persistence. Patients treated with GH in childhood for idiopathic GHD, Turner`s syndrome, Noonan syndrome, Prader Willi syndrome, chronic renal failure or being small for gestational age at birth generally do not need continuation of GHr in adulthood. In patients continuing GHr treatment in the TP, the typical daily replacement doses deflect from the growth-promoting doses of childhood (25–66 µg/kg/d) and diminish towards adult doses. Usually commencing at 0.4–0.5 mg/d, GHr doses are individually titrated in 0.1–0.2 mg intervals every 1–2 months, targeting the upper age-specific reference IGF-1 values and minding clinical response and tolerability [9,10]. Daily GHr doses in transition rarely exceed 1.6 mg/d, with girls typically requiring higher doses than boys.

Due to the lack of anabolic and lipolytic effects of GH, adult GHD patients are clinically characterized by a decrease in muscle and bone mass as well as an increase in abdominal fat mass. These manifestations are present regardless of age, but they are most striking during the transition period [2]. GH and IGF-I along with sex hormones are essential in the period of adolescence and young adulthood and crucially involved in bone remodeling, bone mass accrual and achievement of favorable body composition. A normal GH plasma level is necessary for physical development in late adolescence in order to achieve full maturation of skeletal and muscular systems. Unbalanced body composition (BC), which is a hallmark of COGHD, consequently leads to increased cardiovascular risk and insulin resistance.

COGHD patients aged 16 to 25 years represent a very heterogeneous group, driving often discordant results of studies dedicated to their bone, muscle and fat mass. Various COGHD etiologies, different disease durations and other concomitant hormonal deficiencies and corresponding replacement are all important factors reflecting on bone mass and body composition in that group. Therefore, baseline characteristics and the response to GHr on the skeletal system and body composition are often variable in the TP [13]. Unlike adult-onset GHD, in which a pituitary tumor and other acquired causes are the most frequent causes of GHD, in COGHD, congenital causes predominate in prevalence [14]. In our previous study enrolling 170 patients with COGHD in the transition period, the most prevalent cause of COGHD was congenital impairments (50%), while tumoral and idiopathic GHDs were present in approximately a quarter of the investigated patients [15]. In contrast to our results, other studies have shown that the greatest prevalence of COGHD in young adulthood is attributed to acquired causes or unknown etiology (idiopathic) [16,17,18]. Traumatic brain injury is becoming an increasingly recognized etiological factor of COGHD in the TP, attributable to sports and traffic injuries but possibly also to child abuse-related injuries [19]. The aim of this review article is to highlight the consequences of COGHD after the end of linear growth on bone mass and body composition and the opposing beneficial effects of continuing GH replacement in young adults.

## 2. Physiological Role of GH on Bone, Muscle and Fat Tissue

GH and IGF-I are instrumental regulators of bone metabolism with a pivotal role in longitudinal bone growth and control of bone mass. Apart from direct effects on bone, GH exerts its actions through circulating and local tissue IGF-I. Synergistically with sexual steroids during puberty and young adulthood, these two hormones affect bone modeling and remodeling towards the achievement of adequate PBM [20,21]. This supports the view that the disruption of the GH/IGF-I axis leads to decreased bone formation and a tendency to osteoporosis. The molecular mechanisms behind these effects are complex, weaving a network with other mediators, such as IGF-binding proteins (IGFBPs), PTH and several cytokines (IL1b, IL-6, TNFα) [22]. Bone cells express receptors for both GH and IGF-I, through which GH stimulates the maturation, proliferation and differentiation of chondrocytes and osteoblasts, while IGF-I dominantly reduces osteoblast apoptosis. Besides its direct effects on bone, GH increases renal retention of vitamin D, adding to improved bone mineralization [23]. In order to maintain a stable bone mass and remodeling cycle, GH and IGF-I exert an effect on osteoclasts, leading to bone resorption. IGF-I promotes osteoclast proliferation via the expression of receptor activator of nuclear factor kB ligand (RANK-L), while GH stimulates osteoprotegerin synthesis and, thus, mitigates the osteoclast activity [24,25]. Under the physiological remodeling cycle, bone formation exceeds bone resorption [26]. GHD causes a decreased bone turnover rate, resulting in unbalanced levels of bone formation and resorption markers and, consequently, diminished BMD [21,27,28,29]. Histopathological analyses of bone in adult GHD patients revealed a decreased and eroded bone surface and reduced osteoid compared to healthy controls [30]. Patients affected by GHD demonstrate a mild state of PTH resistance, which recovers during GH treatment [31]. The effect of GH replacement on bone is biphasic. In the first phase, lasting up to one year, bone resorption increases, which may reflect on an initial BMD decrease measured by DXA. However, after at least 1–2 years of GHr, bone formation predominates, with a further increase in bone mass [21,32,33]. Adults with COGHD have a smaller bone size, leading to the possible underestimation of BMD measured by DXA [27]. Most data support the concept that young adults with COGHD who discontinue GHr in the transition period have lower BMD from the predicted genetic potential [34].

GH is a pleiotropic hormone that exerts multiple effects, including those on muscle and adipose tissues. Its strong anabolic action, directly or mediated via IGF-I, is manifested on bone and skeletal muscles, while its catabolic action targets the fat mass [35]. Molecular investigations revealed that GH increases protein synthesis and blocks their degradation, notably in the muscle. Prolonged GH treatment leads to a redistribution of amino acids toward protein formation and blocks their degradation, resulting in an increase in lean body mass (LBM) [36]. Whole body protein turn-over is diminished in adult GHD patients [37]. In addition, GH contributes to controlling the structural properties of muscles, such as muscle fiber composition and fiber type distribution [38,39,40,41]. In this way, GH affects the skeletal muscle contractility and strength.

Besides the effect on muscles, GH, as an insulin antagonist, demonstrates potent metabolic effects, inducing lipolysis. Its dominant lipolytic effect results in an increase in circulating free fatty acids (FFA) and suppression of FFA uptake. Recent reports have underlined the role of GH in complex adipose tissue structure regulation via its impact on adipocyte differentiation and maturation [42]. In humans, GH regulates fat distribution to particular anatomical regions. Contrary to insulin, which causes adipose tissue accumulation, GH accelerates the breakdown of fat mass (FM), primarily in the abdominal compartment [43]. During late adolescence, a peak gain in muscle mass is associated with peak bone mass. In the period of transition to adulthood, GH and IGF-I in conjunction with sex steroids lead to maximal physical development [44].

## 3. Bone Mineral Density (BMD) in COGHD Patients in Transition

### 3.1. BMD at Achievement of near Adult Height

Investigations on the baseline characteristics of bone mass assessment after reaching AH were often incoherent. Bias may be generated by DXA methodology itself. As a surrogate two-dimensional representation of a three-dimensional entity, DXA in short individuals reports lower apparent BMD compared to tall persons with the same actual BMD [45]. Most have reported low bone mineral density (BMD) in young adults with COGHD compared to individuals with normal GH secretion [15]. Some authors have shown that COGHD patients, even if adequately treated with GHr in childhood, demonstrated low BMD after the end of growth. These patients failed to achieve an adequate peak bone mass—defined as the maximum bone mass resulting from the normal growth of an individual [46]. A low PBM is considered a predictor of osteoporosis development and fractures in adulthood. During puberty, approximately 40% of total skeletal mass is accumulated, while PBM is reached by the age of 23 years in males and 20 years in females [3]. In healthy individuals, after reaching AH, accelerated bone acquisition occurs over the next 7 years. The rise in GH and sexual steroids during puberty and late adolescence is crucial to achieve bone apposition. COGHD patients are deprived of adequate skeletal maturation and demonstrate delayed PBM. The lower BMD in adults with COGHD is universally recognized as a consequence of inadequate bone accrual during the TP rather than a later loss of bone mass [47,48]. The achievement of AH occurs several years earlier than the achievement of PBM and peak muscle mass [49]. Underwood et al. found that 80% of 64 investigated COGHD patients (mean age 23 years) had significantly reduced spinal BMD at baseline (near the time of reaching AH), despite having received GHr during childhood [46]. This may reflect dose-related effects of GHr in childhood or late commencement or interruptions in GH treatment. BMD in young adults was reported to significantly correlate with GHr duration prior to reaching AH [50]. Insufficient GHr dosing or discontinuous GHr leads to suboptimal PBM and AH, which positively correlates with BMD value. Unachieved target AH will result in low bone mass, highlighting the imperative of GHr administration in an adequate dose and duration during childhood [51,52]. Most studies demonstrated reduced bone mass at adult height in COGHD patients, while normal BMD at that age of the affected subjects was rarely reported [53,54,55,56,57]. An earlier study found that the smaller bone size reflected on a lower BMD compared with healthy controls [58]. Similarly, Boot et al. reported significantly lower BMD in 40 adolescents with COGHD compared to controls at AH achievement and during the subsequent 2 years [59]. One of the reasons for decreased BMD in this group of patients may be reduced AH. The increased bone fragility described in COGHD was interpreted as a consequence of small bone diameter rather than reduced BMD [60]. In our earlier investigation, we detected Z-sc < −2 at the lumbar spine in approximately 30% of patients adequately treated with GHr in childhood and in 46% of the GH-untreated COGHD group [15]. Kaufman et al. confirmed that patients with COGHD demonstrated a lower bone mass despite regular treatment with GHr in childhood [57]. Likewise, Sagesse et al. reported that approximately 20% of COGHD patients optimally treated with GHr in childhood had a spine BMD between −1 and −2 SD at adult height, reflecting their risk of fractures in later life [50]. Hydtsrup et al. reported that COGHD patients in young adulthood exhibited lower BMD than healthy controls, displaying reduced cortical thickness and overall cortical mineral content [49]. In support of this, an earlier observation reported a more pronounced radial BMD decrease representing cortical bone than in the lumbar spine or hip (30% and 20%, respectively) [57]. On the contrary, others reported that COGHD patients in adolescence have normal BMD up to two years after GHr termination [56]. However, patients that discontinued GHr had biochemical markers indicative of reduced bone remodeling. The results of a recent study contribute to the importance of bone mass assessment after achieving AH. Lange et al. demonstrated that some patients who had recovered the GH/IGF-I axis based on the ITT demonstrated persistently lower BMD than healthy controls [61]. Therefore, they proposed that, as a criterion of persistent GHD in the transition period, the BMD value should be included along with the response to stimulation tests.

In summary, patients with COGHD will have reduced bone mass after completion of longitudinal growth, even if they were on optimal GH replacement during childhood. The reasons for low BMD in this age group are unachieved peak bone mass, small bone size and the often-lower adult height of these subjects compared to healthy controls.

### 3.2. Predictors of BMD at Attainment of Adult Height in COGHD Patients

Longitudinal growth is considered completed when the growth rate is 1 cm/year and after reaching the bone age of 15 years for girls and 17 years for boys. It is assumed that, at this point, 99% of the individual’s AH is achieved [44]. Individuals with COGHD have a lower AH than patients with adult-onset GHD [62,63,64]. Our previous study showed that a longer duration of GHr in pediatric age correlated positively with BMD at the lumbar spine (LS) and femoral neck (FN) in the transition period [15]. Similarly, another report showed that age at GHr treatment cessation before transfer to adult endocrine care correlated negatively with BMD at the LS [65]. KIMS database analysis detected that the gap length between pediatric and adult GHr was negatively associated with bone mineralization at the FN in adults [23]. After transfer from the pediatric clinic at age 18, BMD at the LS and FN was higher in patients receiving GHr during childhood compared to those deprived of this treatment before the TP [15]. Kaufman et al. demonstrated osteopenia both in patients with isolated GHD and patients with multiple pituitary deficiencies (MPHD), supporting the assumption that GH per se affects bone abnormalities [57]. This observation was independently confirmed by MPHD patients having a higher or similar BMD compared with isolated GHD [50]. In the same study, the severity of GHD at diagnosis was not associated with the BMD value past the achievement of AH. KIMS database, investigating 314 COGHD patients, demonstrated that thyrotropin deficiency was also associated with lower BMD at the LS [23].

Approximately 50% of the fractures in adults with isolated GHD developed at normal BMD values. This signals that bone remodeling and bone architecture are both disturbed in the state of low GH levels, reaffirming the view that GH is responsible for skeletal quality [66]. Compared to adult onset GHD, patients with COGHD are generally better responders to GHr, especially from the skeletal system and body composition aspects [67]. The failure of adequate replacement with other missing hormones in addition to GH contributes to BMD worsening [68]. Decreased muscle mass, frequently present in COGHD patients, also contributes to reduced BMD.

In patients developing COGHD as a consequence of intracranial tumors or childhood malignancies, bone health impairment is usually more severe. Cohen et al. investigated BMD in 36 adolescents (mean age 17 years) with a history of a hypothalamic-pituitary tumor treated by surgery, chemotherapy and/or radiotherapy [69]. Childhood cancer survivors are prone to reduced bone mass and delayed bone mineralization. The tumor itself, malignant process as a systemic disease, chemotherapeutic agent characteristics, radiotherapy doses, malnutrition and decreased physical activity all alter bone metabolism and contribute to the risk of reduced BMD [70,71]. All of the above factors may independently affect the GH/IGF-I axis and cause other endocrinopathies, which may additionally deteriorate skeletal status [72,73,74]. It was observed that GHr during childhood contributed to higher Z-sc at the spine and femoral neck in these subjects [69]. We analyzed bone health variables in 142 COGHD patients treated with GHr during childhood at first evaluation upon transfer from pediatric care [15]. Survivors of endocranial tumors and acute lymphoblastic leukemia had significantly lower Z-sc at the lumbar spine than those with congenital or idiopathic GHD (Z-sc values −1.72, −1.34 and −0.95, respectively). Serum bone turnover markers were notably reduced in the tumor-related group.

In summary, possible predictors of higher BMD in patients with COGHD are a longer use of GH treatment during childhood and shorter interruption interval of this therapy between pediatric and adult age. Reduced muscle mass and other endocrinopathies that are common in these individuals correlate negatively with bone mass. Childhood cancer survivors are particularly susceptible to a reduction in bone mineralization due to the tumor as a systemic disease and different antitumor modalities, such as radiotherapy, chemotherapy or immunotherapy. These treatment procedures directly or indirectly impair bone mass and quality.

### 3.3. BMD Dynamics upon GH Replacement in Transition and Young Adulthood

Reports are scarce on original research of GH replacement effects in COGHD patients during the transition period. Inconsistent data are reported from studies dedicated to the effects of continuing or discontinuing GHr after achieving AH (Table 1). Given that GH exhibits important anabolic actions on bone, the discontinuation of GHr at the completion of linear growth induces reduced PBM [75]. Studies conducted twenty years ago described the impact of GHr recommencement in young adults with COGHD. A two-year placebo-controlled study reported a dose-dependent increase in BMD in the LS after 24 months of GHr in COGHD young adults [46]. Other authors have also confirmed that continuation of GHr in the TP is associated with increasing BMD in the lumbar spine by 2 to 6% after 12–24 moths of follow-up [59,76,77,78]. Analyzing 40 young adults with COGHD in the TP after 3 years of GH replacement continuation, we demonstrated a significant BMD increase at the lumbar spine by 7% (Figure 1a,b) [15]. A large study investigating the effects of GHr reinstitution on BMD after adult height achievement in 128 young adults (mean age 19 years) reported no difference after one year of follow-up on BMD and TBMC but significant improvement at 2 years compared to the control [52]. A recent study reconfirmed an increase in hip BMD after 12 months of recommencement of GHr in the TP [79]. In spite of the expected gender differences in response to GHr in young COGHD adults, similar changes in bone mass and bone turnover markers were reported in both sexes [80].

Most investigations focused on the impact of GH recommencement on trabecular bone (lumbar spine) or mixed bone compartments (hip) rather than on the cortical bone. Hyldstrup et al. showed that 24 months of GHr continuation in 160 young adults after AH achievement resulted in significantly increased cortical bone thickness [49]. Cortical bone accrual leads to higher PBM and may reduce the risk of fractures. Some investigations revealed no significant changes after 6–24 months of GHr discontinuation after growth completion compared to control subjects [81] (Table 1). In keeping with that, Cohen et al. did not observe a significant difference in BMD between patients with tumor-related COGHD continuing or discontinuing GHr after 1.7–3.5 years of follow-up [69].

A beneficial effect of GH treatment on bone histomorphology has been reported in the study by Yang et al. They demonstrated a significant improvement in bone microarchitecture induced by six months of GHr in COGHD male adults [82]. An increased risk of fractures in GHD adults has been described in several studies, while there are no published data in patients younger than 30 years [28,83,84]. Vanuga et al. evaluated trabecular bone score (TBS) as a novel method of bone microarchitecture assessment in 63 GHD adults and detected no significant TBS increase during 10 years of follow-up [85]. However, another study examining GHD adults showed a notable TBS increase after 2 years of GHr [68]. The available literature provides no data on TBS analysis in COGHD patients in the transition period.

In summary, the majority of published clinical studies have shown that administration of growth hormone during the transition period significantly increases BMD and total skeletal mass. In support of this, the histomorphological investigations showed a beneficial effect of GHr on bone in COGHD subjects. However, some reports postulated that discontinuation of GH treatment for at least two years will not have negative consequences for the skeletal system. The limitations of such studies are a small number of subjects and a lower dose of GH than usual for late adolescence.

## 4. Body Composition (BC) in COGHD Patients in the Transition Period

### 4.1. BC at Achievement of near Adult Body Height

An unfavorable body composition (BC) is one of the characteristics of COGHD and has implications for bone mass. Most of the investigations reported a decrease in lean body mass (LBM) and an increase in fat mass (FM) in patients with persistent GHD during the TP [86,87,88]. Peak muscle mass in young adulthood correlates with the achievement of PBM and BMD. Low muscle strength and diminished muscular performance were repeatedly reported in COGHD patients [44,89]. At first evaluation of 142 patients after transfer from the pediatric care unit to the adult care unit, we detected that AH and body mass index (BMI) correlated negatively with FM and positively with LBM. In addition, a higher FM was associated with the number of pituitary axes deficiencies [15]. Regarding etiology of COGHD, Yuen et al. found that adults under 30 years treated for craniopharyngioma in childhood had a higher BMI and FM compared to those with other COGHD etiologies. They had more comorbidities and more prevalent hypothalamic syndrome with hyperphagia as well as a higher prevalence of MPHD [90]. Likewise, we detected that tumor-related COGHD was associated in the TP with significantly increased body weight, BMI and waist circumference compared to other etiologies of insufficient GH secretion. Patients with idiopathic COGHD demonstrated a higher LBM and lower FM compared to other etiologies [15]. The prevalence of metabolic syndrome in patients at near the achievement of AH was 15% in patients with cranyopharyngeoma, which is significantly greater than in the age-matched healthy population [91]. Korean authors analyzed 187 COGHD patients in the TP mainly affected by organic causes of GH insufficiency. The subjects investigated during GH treatment discontinuation (2.8 years of interruption) showed a deterioration of the parameters of metabolic syndrome and increased body weight [79]. Another study reported that, upon discontinuation of GHr after the age of 18 years, FM increased progressively in parallel with a decline in LBM and subsequent reduction in muscle strength and exercise performance [86].

Binder et al. analyzed BC in adolescents after six months of GHr cessation at near achievement of AH and reported a gain in FM and loss in LBM in those with severe GHD after retesting in the TP [92]. An increased FM compared to healthy controls was reported in five severe GHD patients with a GHRH receptor mutation who were GH treatment naïve. Two years of GHr resulted in normalization of adipose tissue mass in all five investigated subjects [93]. Another study assessed BC and muscle strength in 18 young males with COGHD treated with GHr until the achievement of AH. Increased total FM and compartmental truncal, limb and abdominal FM were reported compared to healthy individuals along with lower total LB and truncal and limb LBM [44].

In summary, the lack of adequate GH secretion will result in the absence of its lipolytic and anabolic effects. Consequently, such individuals will have an increased body FM and decreased muscle mass. These are characteristics of all age groups with GHD, but the sequelae on body composition are most noticeable during the transition period. Peak muscle mass will be absent if sufficient peak bone mass has not been achieved. Regarding body composition, the etiology of GHD in childhood plays an important role. Patients with a childhood history of craniopharyngioma have a higher prevalence of metabolic syndrome in young adulthood compared to age-matched healthy controls as well as increased FM than other COGHD etiologies. On the contrary, idiopathic COGHD patients show higher LBM compared to a congenital or tumor-caused absence of GH secretion.

### 4.2. Changes in BC on GH Replacement in Transition and Young Adulthood

Similar to the discordant reports related to bone mass, the findings on the impact of continuing GHr on body composition in the TP are equally incoherent (Table 2). Our investigation observed the favorable effects of the recommencement of GHr in 40 COGHD subjects after 3 years in the TP (Figure 2) [15]. Likewise, other authors showed that reinstitution of GHr in the TP with a duration of 12–24 months induced a significant LBM increase (6–14%) and FM reduction (7–12%) [46,94,95]. Cessation of GHr after achievement of AH for 6–24 months in COGHD adolescents resulted in an LBM decrease by up to 8% and FM increase up to 17% [96,97,98]. Over a period of 24 months following discontinuation of GHr in 40 older adolescents, a significant FM increase and LBM decrease by 4.5 kg were detected, testifying to the dual adverse effects of cessation of GH treatment in the TP [96]. The favorable impact of a two-year continuation of GHr on body composition during the TP was reported by an investigation involving 149 patients, with effects more pronounced in males [51] (Table 2). The observed gender differences were expected considering the corresponding body composition distinctions between genders in healthy individuals. A dose dependency of GHr effects on BC was reported by some authors but not by others. More favorable changes in BC are expected on higher GHr doses [46]. Vhal et al. reported that GHr cessation resulted in unbalanced changes in FM and LBM, which improved during two years after GHr resumption [99]. GH and IGF-I promote an increase in muscle mass, which positively affects muscle strength and contractility [100,101]. However, a few studies reported opposite results [81,102]. Camtosun et al. found no differences in BC between patients who remained GHD after completion of growth and those who recovered the GH/IGF-I axis at baseline and after 6 months of GHr cessation [102]. Mauras et al. observed no difference in LBM and FM after 24 months of GHr discontinuation but reported a trend of improvement of BC during the first 12 months, which was lost by the 2 years of follow-up [81]. Some authors explained the absence of GHr effects on body composition by a younger patient age than in comparative studies, higher prevalence of idiopathic GHD and failure to recognize BC alterations in periods shorter than 2 years [45]. Favorable body composition dynamics were reported not only on standard daily GHr but also with the novel long-acting GHr formulations [103,104,105].

In summary, besides the beneficial effect of GH replacement on the skeletal system during the TP, the improvement of body composition has been reported by numerous clinical studies. The normalization of IGF-I levels by GH therapy leads to a significant increase in muscle mass and, in parallel, to a reduction in fat mass, mainly in the abdominal region. However, some authors dispute the effect of GHr on lean body mass and fat mass, postulating that stopping GH therapy in the TP will not affect body composition. Regarding BC, a superior response to GHr may be expected in males and those on higher doses of GH.

## 5. Conclusions

This review describes the impact of COGHD and growth hormone replacement on bone mass and body composition in the transition period and young adulthood. After more than three decades of accumulated experience in growth hormone replacement after completion of growth, the clinical data and published studies confirm numerous benefits on physical and mental health. A prime target group for the implementation of GH treatment is patients in the period of transition and young adulthood. One of the most important effects of GH replacement is facilitation in the reach of adequate peak bone mass, which is a crucial determinant for the achievement of normal bone mass and reduction in fracture risk in later life. Therefore, GH replacement should not be discontinued at the end of longitudinal body growth when bone mass is still accruing. Bone mineral density and body composition analysis must be made mandatory components of baseline and follow-up assessments for GHD patients in transition to young adulthood. Based on the majority of prospective trials, including our investigations, we advise the continuation of GH replacement in the transition period and young adulthood in order to achieve normal bone mass and favorable body composition as important predictors of better adulthood health outcomes.

## Figures and Tables

**Figure 1 ijms-25-10313-f001:**
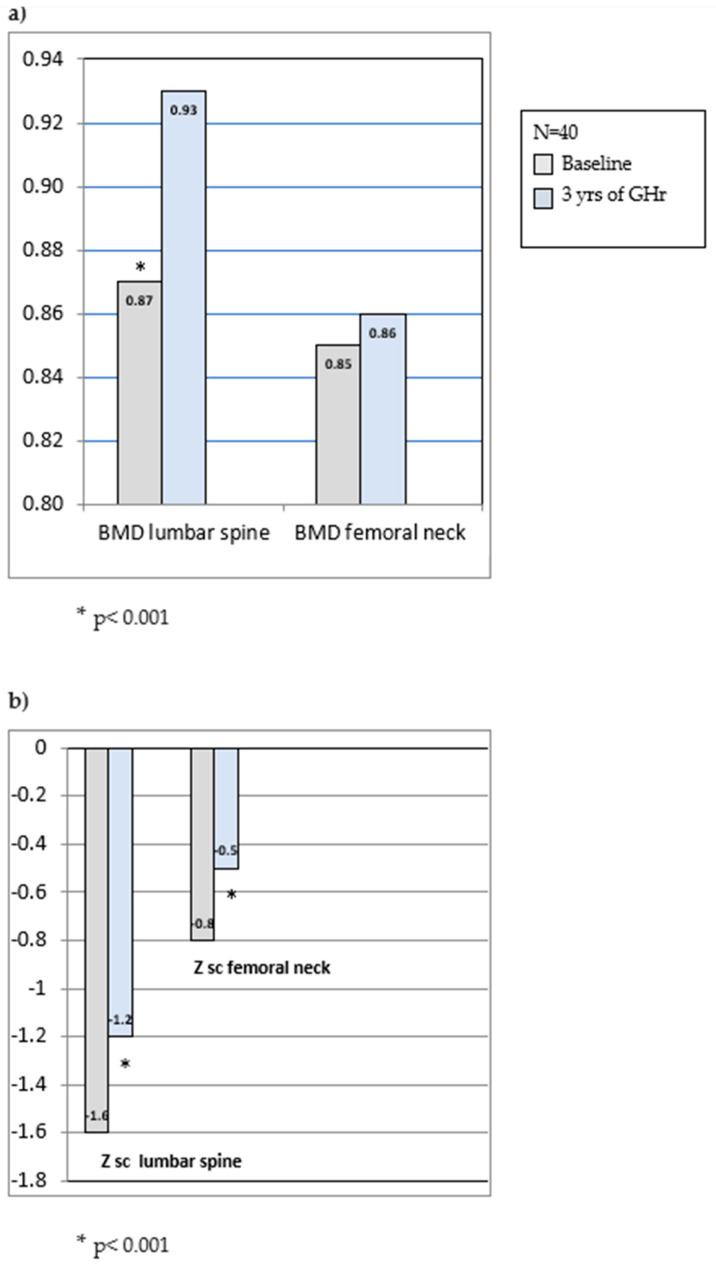
(**a**) Changes in bone mineral density (BMD) at the lumbar spine and femoral neck upon completion of growth (baseline) and after 3 years of GH replacement in the transition period; (**b**) Z sc at the lumbar spine and femoral neck upon completion of growth (baseline) and after 3 years of GH replacement in the transition period (n = 40, mean age 18.8 ± 2.0 years, range 16–25 years) Based on [16].

**Figure 2 ijms-25-10313-f002:**
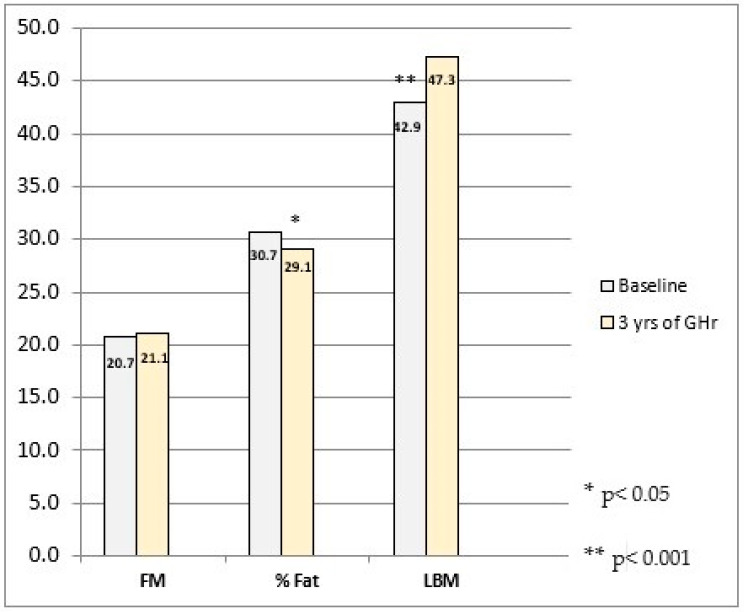
Changes in body composition (fat mass—FM, fat percentage—%F and lean body mass—LBM) upon completion of growth (baseline) and after 3 years of GH replacement in the transition period (n = 40, mean age 18.8 ± 2.0 years, range 16–25 years) Based on [16].

**Table 1 ijms-25-10313-t001:** Changes in bone mineral density in COGHD patients on GH replacement during the transition period.

Study	N of Subjects	Age of Subjects (Years)	Etiology of COGHD	Duration of GHr (Months)	Dose of GHr	Changes in Spine BMD (g/cm^2^)	Changes in Spine BMD (Z Score)	Changes in FN BMD (Z Score)	Changes in Total BMD (Z Score)	Changes in TBMC (%)	Changes in Cortical Bone (%)
[46] Underwood et al., 2003.	64	23.8 ± 4.2	27 intracranial tumor 23 idiopathic MPHD 12 isolated GHD	24	Placebo 12.5 μg/kg/d 25 μg/kg/d	+1.3% +3.3% +5.2% *p* ˂ 0.01	/	/	/	/	/
[75] Drake et al., 2003.	24	17.0 ± 1.4	9 intracranial tumor 7 idiopathic MPHD 4 isolated GHD 2 congenital 2 ALL	12	Controls (untreated) 0.35 IU/kg/wk	+2.6% +4.7% *p* ˂ 0.01	/	/	/	+2.4 +6.1 *p* > 0.05	/
[81] Mauras et al., 2005.	58	15.8 ± 1.8	6 idiopathic MPHD 10 isolated GHD 1 congenital 1 ALL 5 CNS radiation 6 other	24	Placebo 20 μg/kg/d Controls (recovered GH axis)	/	−1.08 −0.29 −0.23 *p* > 0.05	/	/	/	/
[59] Boot et al., 2009.	29	Adolescents at adult height (not specified)	8 tumor related 12 structural abnormalities 9 idiopathic GHD	24	Controls (untreated) 0.7 mg/d	/	−0.69 −0.9 *p* > 0.05	/	−1.11 −0.55 *p* > 0.05	/	/
[58] Conway et al., 2009.	160	21.1 ± 2.3	Not specified	24	Controls (untreated) 0.6–1.4 mg/d	2% 6% *p* ˂ 0.01	/	0.0 0.02 *p* ˂ 0.01	/	/	/
[69] Cohen et al., 2012.	36	16.9 ± 1.9	Brain tumor	27	Controls (untreated) Dose not specified	*p* > 0.05	/	*p* > 0.05	/	/	/
[49] Hyldstrup et al., 2012.	161	21.2 ± 1.9	Not specified	24	Controls (untreated) 0.6–1.4 mg/d	/	/	/	/	/	*p* > 0.05 +6.4 *p* ˂ 0.01
[15] Doknic et al., 2021.	40	19.2 ± 2.0	15 tumor related 2 ALL 22 congenital 1 idiopathic	36	0.5 mg/d	+6.9% *p* ˂ 0.01	*p* ˂ 0.01	+1.2% (g/cm^2^) *p* > 0.05 Z sc *p* ˂ 0.01	/	+8.8 *p* ˂ 0.01	/
[79] Lee et al., 2022.	187	20.0 ± 3.0	146 intracranial tumor 21 congenital 5 infiltrative 10 cranial irradiation 5 idiopathic GHD	12	0.4 mg/d	/	−0.10 *p* > 0.05	+0.2 *p* ˂ 0.05	/	/	/

MPHD—multiple pituitary hormone deficiency; GHr—growth hormone replacement; BMD—bone mineral density; FN—femoral neck; TBMC—total bone mineral content; DXA (Dual-Energy X-ray Absorptiometry) method was used for evaluation of body composition.

**Table 2 ijms-25-10313-t002:** Changes in body composition in COGHD patients on GH replacement during the transition period. MPHD—multiple pituitary hormone deficiency; GHr—growth hormone replacement; FM—fat mass; LBM—lean body mass; DXA (Dual-Energy X-ray Absorptiometry) method was used for evaluation of body composition.

Study	N of Subjects	Age of Subjects (Years)	Etiology of COGHD	Duration of GHr (Months)	Dose of GHr	Changes in FM	Changes in LBM	Changes in BMI
[99] Vahl et al., 2002.	19	20.2 ± 0.65	6 intracranial tumor 13 idiopathic GHD	24	Placebo 0.5–2.0 IU/m^2^/day	+3.8 kg −1.6 kg *p* > 0.05	+1.8 kg +9.8 kg *p* ˂ 0.001	/
[46] Underwood et al., 2003.	64	23.8 ± 4.2	27 intracranial tumor 23 idiopathic MPHD 12 isolated GHD	24	Placebo 12.5 μg/kg/d 25.0 μg/kg/d	+2.8% −3.8% −7.7% *p* ˂ 0.001	−2.0% +3.3% +6.6% *p* ˂ 0.001	/
[51] Attanasio et al., 2004.	149	19.6 ± 2.8	Isolated GHD and MPHD	24	Control (untreated) 12.5 μg/kg/d 25.0 μg/kg/d *p* ˂ 0.001	12.9% −7.1% −6.0%	+ 2.4% +12.7% +14.2% *p* ˂ 0.001	/
[94] Carroll et al., 2004.	24	17.0 ± 0.3	9 intracranial tumor 2 ALL 1 congenital 12 idiopathic	12	Control (untreated) 0.35 U/kg/wk	−1.4% −2.7% *p* > 0.05	−2.0% +6.0% *p* ˂ 0.001	/
[81] Mauras et al., 2005.	58	15.8 ± 1.8	6 idiopathic MPHD 10 isolated GHD 1 congenital 1 ALL 5 CNS radiation 6 other	24	Placebo 20 μg/kg/d Controls	+4.7% +5.5% +3.8% *p* > 0.05	−5.5% −5.4% +4.0% *p* > 0.05	/
[59] Boot et al., 2009.	29	Adolescents at adult height specified	8 tumor related 12 structural abnormalities 9 idiopathic GHD	24	Controls (untreated) 0.7 mg/d	0.50 (Z sc) 0.71 (Z sc) *p* > 0.05	−2.05 (Z sc) −0.93 (Z sc) *p* ˂ 0.05	/
[98] Bazarra- Castro et al., 2012.	75	16–25	20 idiopathic GHD 55 organic GHD	/	23 months off GHr	/	/	+3.5 kg *p* ˂ 0.05
[92] Binder et al., 2019.	90	16.7 ± 1.1	78 isolated GHD 12 MPHD 10 pituitary malformation	/	6 months off GHr: Severe GHD Non-severe GHD	+9.0% +4.0% *p* = 0.06	−10.0% −3.0% *p* ˂ 0.05	/
[15] Doknic et al., 2021.	40	19.2 ± 2.0	15 tumor related 2 ALL 22 congenital 1 idiopathic	36	0.5 mg/d	−5.2% *p* ˂ 0.05	+12.0% *p* ˂ 0.001	/
[79] Lee et al., 2022.	187	20.0 ± 3.0	146 intracranial tumor 21 congenital 5 infiltrative 10 cranial irradiation 5 idiopathic GHD	12	32 months off GHr 12 months on GHr 0.4 mg/d	/	/	+1.8 kg +0.5 kg *p* ˂ 0.001

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
