# Peer review of "Transition Period and Young Adulthood in Patients with Childhood Onset Growth Hormone Deficiency (COGHD): Impact of Growth Hormone Replacement on Bone Mass and Body Composition"

_ijms, 2024, doi:10.3390/ijms251910313_

Round 1

Reviewer 1 Report

Comments and Suggestions for Authors

Doknic and colleagues summarize the extant data for changes in bone and body composition at the transition phase for subjects with childhood onset GHD of multiple etiologies. They report multiple studies, but do not at the end of each section summarize the data and give their interpretation of these sometimes-discordant studies. The authors are the experts and should note their opinion. It seems as if each study is equally valid, even if discordant. Thus, the requirement for their interpretation.

1. The data are presented in paragraph format making it difficult to follow what was done. Extensive tables would greatly help this manuscript noting as summaries of previous studies: number of subjects, etiology of GHD, GH dose and duration, method of body composition determination and age at last evaluation.

2. abstract something should be noted about dose of hGH

3. L. 4 and ff (multiple times) Please use adult height (AH) or near AH rather than final body height (FBH) for the latter has a much different meaning

4. age of peak bone mass differs within the manuscript. Please note the one that you believe best with a proper reference.

5. BMD is a 3 dimensional property based on a 2 D dexa which i why short persons have a lower apparent BMD than tall persons at the same actual BMD. This concept should be stated in a paragraph with proper references.

6. Another paragraph should state how GHD is diagnosed at near adult height to decide whether that individual will be GHD as an adult. The notion of how to dose, that is transition from the pediatric dose to an adult dose should also be discussed. Add a sentence or two that all of the other indications for childhood GH treatment with the possible exception of Prader-Willi syndrome do not require retesting or continuing GH treatment. It should be noted that the vast majority of those with a diagnosis of GHD will not require additional GH therapy throughout the transition period and as adults.

Comments on the Quality of English Language

P. 2 5 lines from bottom "mass"

P. 4 3 lines from bottom use "FBH", but note from above that this concept is incorrect

P. 5 L. 1 "criterion"

Author Response

Authors express their sincere gratitude to the reviewers` time and effort and for all valuable comments which will invariably improve the quality of the manuscript.

In response to specific comments:

Reviewer 1

Doknic and colleagues summarize the extant data for changes in bone and body composition at the transition phase for subjects with childhood onset GHD of multiple etiologies. They report multiple studies, but do not at the end of each section summarize the data and give their interpretation of these sometimes-discordant studies. The authors are the experts and should note their opinion. It seems as if each study is equally valid, even if discordant. Thus, the requirement for their interpretation.

A summary of data is added after each section to provide expert interpretation of the presented data.

  1. The data are presented in paragraph format making it difficult to follow what was done. Extensive tables would greatly help this manuscript noting as summaries of previous studies: number of subjects, etiology of GHD, GH dose and duration, method of body composition determination and age at last evaluation.

In order to facilitate the reader`s following of the presented data, tables are now added (Table 1, Table 2) summarizing the most important previous studies in the field in regards of number of subjects, etiology of GHD, GH dose and duration, and the significance of the effect of GHr on BMD and body composition. In all cited studies in the manuscript, method of body composition and BMD determination was DXA, so it is added in the text below Tables 1 and 2. “Age at least evaluation” is not added in Tables 1 and 2, because this data is not specifically mentioned in the cited studies in the manuscript.

  1. Abstract something should be noted about dose of hGH

Information on dose of hGH is added in the Abstract.

  1. L. 4 and ff (multiple times) Please use adult height (AH) or near AH rather than final body height (FBH) for the latter has a much different meaning

The term final body height (FBH) is replaced by the term adult height (AH) throughout the manuscript.  

  1. Age of peak bone mass differs within the manuscript. Please note the one that you believe best with a proper reference.

A single source was selected to best define the age of peak bone mass which is now uniform throughout the manuscript.  

  1. BMD is a 3 dimensional property based on a 2 D dexa which is why short persons have a lower apparent BMD than tall persons at the same actual BMD. This concept should be stated in a paragraph with proper references.

An explanation is added regarding the possible methodological bias caused by DXA being a surrogate areal measurement of a volumetric property (ref 46).

  1. Another paragraph should state how GHD is diagnosed at near adult height to decide whether that individual will be GHD as an adult. The notion of how to dose, that is transition from the pediatric dose to an adult dose should also be discussed. Add a sentence or two that all of the other indications for childhood GH treatment with the possible exception of Prader-Willi syndrome do not require retesting or continuing GH treatment. It should be noted that the vast majority of those with a diagnosis of GHD will not require additional GH therapy throughout the transition period and as adults.

A paragraph is added to explain in brief protocols for reassessing GH sufficiency upon transition to young adulthood, the stratification of probability of GHD persistence based on initial etiology, and on the notion of transition of GH replacement doses from the pediatric to the adult ranges.

Comments on the Quality of English Language

The language errors observed by the reviewer were gratefully acknowledged and corrected and language was edited throughout the manuscript.  

Reviewer 2 Report

Comments and Suggestions for Authors

Dear author, the review is comprehensive and approaches the multiple aspects of the subject in an organised and understandable manner. The studies considered were all relevant and contributed to the scientific soundness of the work. The aim and implications drawn are relevant, they provide a significant contribution for clinical advancement in the treatment of COGHD patients with a great potential for enhancing their health and quality of life. Improvements could be made to the abstract namely mentioning first the aim of the study and only after that the conclusions reached to avoid confusion.

Author Response

Authors express their sincere gratitude to the reviewers` time and effort and for all valuable comments which will invariably improve the quality of the manuscript.

In response to specific comments:

Reviewer 2

Dear author, the review is comprehensive and approaches the multiple aspects of the subject in an organised and understandable manner. The studies considered were all relevant and contributed to the scientific soundness of the work. The aim and implications drawn are relevant, they provide a significant contribution for clinical advancement in the treatment of COGHD patients with a great potential for enhancing their health and quality of life. Improvements could be made to the abstract namely mentioning first the aim of the study and only after that the conclusions reached to avoid confusion.

Abstract is corrected and the aim of the study is expressed initially, followed by conclusions.

Round 2

Reviewer 1 Report

Comments and Suggestions for Authors

The investigators have responded to my original comments with short summaries at the end of individual sections as well as constructing two tables to make the results of studies previously performed more accessible to the reader. 

1. page 2/23 more appropriate might be 25 to 66 and not 100 micrograms per kg per day.

2. Page 9/23 first paragraph "increase" rather than improve for a parameter

3. The tables are well constructed. The authors should go back to the written material to try to reduce the now redundancies between the prose and the tables.

Comments on the Quality of English Language

This is improved but there are still some articles (a, the) missing before some nouns

Author Response

The authors are grateful for reviewer`s comments and have implemented the requested alterations.

Reviewer 1

The investigators have responded to my original comments with short summaries at the end of individual sections as well as constructing two tables to make the results of studies previously performed more accessible to the reader. 

  1. page 2/23 more appropriate might be 25 to 66 and not 100 micrograms per kg per day.

This is corrected accordingly.

  1. Page 9/23 first paragraph "increase" rather than improve for a parameter

This is corrected accordingly.

  1. The tables are well constructed. The authors should go back to the written material to try to reduce the now redundancies between the prose and the tables.

The manuscript is revised and made more compact with the exclusion of redundancies regarding the material presented in the tables.

Thank you and kind regards,

Mirjana Doknic MD, PhD, Professor

Neuroendocrine Department, Clinic for Endocrinology, Diabetes and Metabolic Diseases; University Clinical Center of Serbia; Faculty of Medicine, University of Belgrade